# Trait reactance and trust in doctors as predictors of vaccination behavior, vaccine attitudes, and use of complementary and alternative medicine in parents of young children

**Anna Soveri**[1]\*, **Linda C. Karlsson**[2], **Otto Mäki**[2], **Jan Antfolk**[2], **Otto Waris**[2,3,4], **Hasse Karlsson**[1,5,6], **Linnea Karlsson**[1,6,7], **Mikael Lindfelt**[8], **Stephan Lewandowsky**[9,10]

**1** FinnBrain Birth Cohort Study, Institute of Clinical Medicine, University of Turku, Turku, Finland, **2** Department of Psychology, Åbo Akademi University, Turku, Finland, **3** Department of Child Psychiatry, Research Centre for Child Psychiatry, University of Turku, Turku, Finland, **4** INVEST Research Flagship, University of Turku, Turku, Finland, **5** Department of Psychiatry, Turku University Hospital and University of Turku, Turku, Finland, **6** Centre for Population Health Research, University of Turku and Turku University Hospital, Turku, Finland, **7** Department of Child Psychiatry, Turku University Hospital and University of Turku, Turku, Finland, **8** Department of Theological Ethics, Åbo Akademi University, Turku, Finland, **9** School of Psychological Science, University of Bristol, Bristol, United Kingdom, **10** School of Psychological Science, University of Western Australia, Perth, Australia

\* anjoso@utu.fi

**Data Availability Statement:** Due to Finnish federal legislation on personal data protection in medical research, the original research data cannot

## Abstract

### Objective

The aim of the present study was to investigate whether anti-vaccination attitudes and behavior, and positive attitudes to complementary and alternative medicine (CAM), are driven by trait reactance and a distrust in medical doctors.

### Methods

The sample consisted of 770 Finnish parents who filled out an online survey. Structural equation modeling (SEM) was used to examine if trait reactance plays a role in vaccination decisions, vaccine attitudes, and in the use of CAM, and whether that relationship is mediated by trust in medical doctors.

### Results

Parents with higher trait reactance had lower trust in doctors, more negative attitudes to vaccines, a higher likelihood of not accepting vaccines for their children and themselves, and a higher likelihood to use CAM treatments that are not included in evidence-based medicine. Our analyses also revealed associations between vaccination behavior and CAM use and vaccine attitudes and CAM use, but there was no support for the previous notion that these associations would be explained by trait reactance and trust in doctors.

be made available online, but data can potentially be shared with Material Transfer Agreement. Requests and collaboration initiatives can be directed to the Board of the FinnBrain Birth Cohort Study. Please contact data manager Teemu Kemppainen (teekem@utu.fi).

**Funding:** AS was funded by the Academy of Finland (grant number: 316004; www.aka.fi/en/). LCK was funded by the Department of Psychology (www.abo.fi/en/study-subject/psychology/) and the doctoral network of Minority Research (www.abo.fi/en/minority-research/) at Åbo Akademi University. ML was funded by the Academy of Finland (grant number: 316726; www.aka.fi/en/) and the Polin Institute (www. polininstitutet.fi/en/ polin-institute/). The funders had no role in study design, data collection and analysis, decision to publish, or preparation of the manuscript.

**Competing interests:** The authors have declared that no competing interests exist.

## Conclusions

Taken together, higher trait reactance seems to be relevant for attitudes and behaviors that go against conventional medicine, because trait reactance is connected to a distrust in medical doctors. Our findings also suggest that high trait reactance and low trust in doctors function differently for different people: For some individuals they might be associated with anti-vaccination attitudes and behavior, while for others they might be related to CAM use. We speculate that this is because people differ in what is important to them, leading them to react against different aspects of conventional medicine.

## Introduction

Vaccination is widely regarded as one of the most important public health achievements. Thanks to successful immunization programs, many serious and highly contagious diseases have become rare and, in some cases, eliminated and even eradicated [1]. Despite the unquestionable benefits of vaccination, previous research has shown that many individuals have concerns about accepting vaccines for themselves or for their children, and some individuals choose to delay or reject vaccinations altogether [2–6]. This phenomenon, labeled vaccine hesitancy [7], poses a threat to global health, as it undermines vaccination coverage and can lead to outbreaks of vaccine-preventable diseases (see [8], for figures on measles outbreaks). Why then do some individuals hesitate in their decision to get vaccinated, or refuse vaccinations altogether, despite the medical consensus about the safety and benefits of vaccines, and the risks of not getting vaccinated?

The results from a number of studies show that the decision to get vaccinated is a complex process that can be influenced by a wide range of factors (for reviews, see e.g., [5,9–15]). Studies that aim at identifying key determinants of vaccination decision-making, suggest that vaccine acceptance is more likely among individuals who perceive vaccines as available, affordable, beneficial, safe, and effective, and who trust the actors involved in vaccinations [7,15,16]. However, even though the relationship between people's vaccine attitudes and their vaccination behavior has received a lot of research interest, the questions of why some, but not all, individuals perceive vaccines negatively, has not been studied as extensively and systematically. Hence, an awareness of which factors influence people's vaccine attitudes is important, for example, when designing interventions that address negative vaccination attitudes. This is because it is the "underlying fears, identity issues and worldviews that motivate people to embrace the surface attitudes" (17; p. 308). Therefore, attempts to increase vaccination uptake may be inefficient if these underlying factors are not properly considered and addressed [17].

One way to think about vaccine attitudes is that they reflect an individual's tendency to agree with the medical consensus that approved vaccines are safe and beneficial. Hence, considering vaccines to be unnecessary or unsafe, means having opinions or beliefs that go against a medical consensus. To investigate this idea further, one line of research has focused on exploring if individuals embrace negative attitudes to vaccines because they have a general unwillingness to accept scientific evidence. One of the first studies on this topic [18] examined the association between conspiratorial thinking, political worldviews, and attitudes to vaccines in a sample of 1001 adults in the U.S.. The study showed that negative attitudes to vaccination were related to a higher tendency for conspiratorial thinking. According to the authors of that study, conspiratorial ideation stands in direct opposition to scientific reasoning, which may

explain why individuals with a tendency for conspiratorial thinking would be motivated to reject scientific evidence that challenges their beliefs [18]. Conspiratorial thinking was also found to be an important predictor of parental vaccination decisions in a recent survey of 4010 U.S. adults [19]. In that study, parents with higher levels of conspiratorial thinking were more likely to have delayed vaccines for their children. A higher belief in conspiracy theories has recently been shown to be related to negative vaccine attitudes also among 518 U.S. adults [20] as well as in a sample of adults from 24 countries [17]. In the latter study, a higher belief in conspiracy theories was shown to be related to more negative attitudes to vaccines in all countries.

Another suggested predictor of opposition to vaccinations is reactance. Reactance refers to the motivational state that arises when people feel that their behavioral freedom has been threatened or taken away [21]. When this occurs, individuals may act contrary to the pre-scribed action in order to protect or restore their feeling of freedom and control. In a recent study, Hornsey et al. [17] found that individuals with higher trait reactance were more likely to reject vaccinations. Trait reactance refers to an individual's predisposition to perceive situations as threats to his/her freedom and to act with reactance (for an overview of trait reactance, see e.g., [22]). The role of reactance in the vaccination context is unsurprising because national immunization programs, the medical consensus around the benefits and safety of vaccines, and the fact that accepting vaccines is considered the norm, may be perceived as threats to people's freedom of choice. Reactance may manifest itself as negative attitudes towards vaccines and medical authorities, and in some individuals, even in a behavior that favors the option that they feel has been taken away from them, that is, to postpone vaccinations or to not get vacci-nated altogether. Therefore, if reactance is the motive behind the negative perceptions of vac-cines, educational interventions by health authorities, which represent one of the most widely used methods to counter negative attitudes to vaccines, may prove inefficient (for reviews on interventions, see e.g., [9,23,24]). Attempts at improving vaccine-related knowledge and cor-recting misperceptions about vaccinations by presenting scientific evidence, may in fact back-fire and result in even stronger negative attitudes to vaccines [25,26]. Reactance may thus undermine the efficiency of educational interventions.

Other studies have looked at the relationship between vaccine attitudes and the unwilling-ness to agree with the medical consensus from a different angle; namely, from the perspective of complementary and alternative medicine (CAM). In an Australian study with adults, Browne et al. [27] showed that negative attitudes to vaccines were associated with a tendency to prefer CAM over conventional medicine. On the basis of those results, and the fact that CAM refers to treatments and substances that are not included in evidence-based medicine, Browne et al. [27] speculated that negative attitudes to vaccines might be related to a reluctance to accept conventional medicine, and to a distrust in authorities providing that kind of evi-dence. This speculation received support in a later qualitative study [28] with 29 Australian parents who had rejected or postponed vaccines. For many of the parents, CAM was consid-ered a natural way to strengthen the immune system, whereas vaccines were considered toxic and harmful. Many of the parents who reported using CAM also mentioned the importance of trusting one's own expertise in knowing what is best for his/her own children. Finally, for many of the parents, CAM also represented an expert system that is free from the influence of "Big Pharma" and that stands in opposition to conventional medical epistemology. The rela-tionship between vaccine hesitancy, use of CAM, trust in CAM, and trust in conventional treatments was recently investigated in 5,200 Spanish adults [29]. Even though the results showed that more CAM use was associated with greater vaccine hesitancy, a distrust in con-ventional treatments played a more important role in explaining vaccine hesitancy than did trust in CAM. Based on this, the authors speculated that people do not become vaccine hesi-tant because they trust CAM, but rather because they distrust conventional medicine. The

connection between positive attitudes to CAM and negative attitudes to vaccines has recently been found also among parents in 18 European countries [30], and adults living in America [31]. Finally, the results from an Australian study with 2758 adults [32], indicated that the negative association between CAM and vaccine attitudes could largely be explained by magical beliefs about health, which lends support to the idea that negative attitudes to vaccinations, as well as CAM, may be due to an underlying view on health that is not evidence-based.

In the present study, we wanted to shed more light on the role of trait reactance and trust in medical doctors in the vaccination context in parents of young children. This population is highly relevant when studying vaccination acceptance, because decisions about vaccinations are of immediate importance for this group. However, instead of looking only at attitudes, which was the focus of the study by Hornsey et al. [17], we explored vaccination behavior as well, that is, whether the parents had accepted childhood vaccines for their children and influenza vaccines for themselves. The second aim of the present study was to investigate the role of trait reactance and trust in medical doctors in predicting parents' use of CAM. Previous studies have suggested that both negative attitudes to vaccines and positive attitudes to CAM may be due to an underlying unscientific view on health and a reluctance to adhere to evidence-based medicine [27,28,32]. The present study tests these speculations in the following two ways: 1) by exploring to what degree trait reactance and trust in medical doctors predict anti-vaccination attitudes and behavior, and CAM use, and 2) by investigating if the association between anti-vaccination attitudes and behavior and CAM use, can be explained by trait reactance and trust in medical doctors. The assumption that trait reactance plays a role also in the decision to use CAM, is based on the idea that CAM represents nonconventional treatments that fall outside the prevailing medical recommendations. Reactance may thus manifest itself in use of CAM in individuals who experience conventional medicine as a threat to their freedom of choice. To the best of our knowledge, this is the first study to look at actual vaccination behavior in this context.

We used structural equation modeling (SEM) to examine if trait reactance predicted vaccination behavior (accepting influenza vaccines for oneself and childhood vaccines for one's children), vaccine attitudes, and CAM use, and whether these relationships were fully or partially mediated through trust in doctors. Because previous studies have suggested that negative attitudes to vaccines and positive attitudes to CAM are related to reluctance in accepting conventional medicine [27–29,32], we hypothesized that higher trait reactance would predict lower trust in doctors, more negative attitudes to vaccines, a lower likelihood of accepting vaccines and a higher likelihood of using CAM. Finally, we tested the hypothesis that trait reactance and trust in doctors would explain some of the association between the vaccine-related outcomes and CAM use.

## Materials and methods

### Study context

In Finland, childhood vaccinations are administered free of charge at child health clinics in accordance with the national vaccination program [33]. The influenza vaccines are included in the national vaccination program free of charge for all risk groups, including children under the age of 7 years. All vaccinations are voluntary.

### Participants and procedure

An invitation to participate in a 20-minute electronic survey was sent out per mail to 3401 Finnish parents participating in the FinnBrain Birth Cohort Study (hereafter called Finnbrain), which is an ongoing longitudinal project investigating child development [34]. All parents

who received the invitation were caregivers to at least one child younger than 4.5 years. In all, 833 parents responded to the survey, but for 50 of them, informed consent was missing, and 13 indicated that they did not allow their responses to be connected to previously gathered data. These individuals were excluded, resulting in a sample of 770 parents (response rate 22.6%; Table 1). Their mean age was 36.43 years ($SD$ = 4.87, range = 22–61). In 155 cases, both parents of the same child had answered the survey.

### Ethics statement

The study received ethical permission by the Ethics Committee of the Hospital District of Southwest Finland. In the invitation letter, the parents received information about the study and that they could terminate their participation at any time. All parents were asked to give their informed consent to participate and to indicate whether they allowed their responses to be connected to their personal data previously collected in the project.

### Measures

The survey was administered in either Finnish or Swedish, depending on the preference of the participant. The measures included in the current study are described below. See S1–S3 Questionnaires for the questionnaires in English, Swedish, and Finnish.

**Childhood vaccination behavior.**   The following three questions queried parents' past vaccination behavior concerning their children's childhood vaccinations: 1) Have you ever hesitated in letting your child(ren) receive any of the childhood vaccines?, 2) Have you ever postponed a childhood vaccination for your child(ren)?, and 3) Have you ever decided not to let your child(ren) receive any of the childhood vaccines? The parents could answer either "yes" or "no" to each question. These questions were combined into a single measure of childhood vaccination behavior as follows: 0 = had never hesitated in a childhood vaccination decision, or postponed or rejected a childhood vaccine, 1 = had hesitated or postponed, but not rejected, a childhood vaccine, 2 = had rejected a childhood vaccine. The response was coded as 0 if the child had medical contraindications for vaccination.

The parents were informed that the term "childhood vaccines" referred to the vaccines included in the national vaccination program for children up to the age of six: the rotavirus vaccine, the chickenpox vaccine, the pneumococcal conjugate vaccine (PCV), the DTaP-IPV-Hib ("5-in-1") vaccine, the MMR vaccine, and the DtaP-IPV ("4-in-1") vaccine.

**Table 1. Descriptive information about the participants.**

| Variable | n | % |
|---|---|---|
| **Sex** | | |
| Female | 500 | 64.94 |
| Male | 270 | 35.06 |
| **Language** | | |
| Finnish | 648 | 84.16 |
| Swedish | 122 | 15.84 |
| **Education**[a] | | |
| Basic/Upper secondary | 180 | 23.38 |
| University of applied sciences | 216 | 28.05 |
| University | 288 | 37.40 |

[a]$n$ = 684.

**Influenza vaccination behavior.** To get a measure of influenza vaccination behavior, the parents were asked whether they had taken the influenza vaccine for themselves during the preceding influenza season. The response alternatives were coded as: 0 = had received the vaccine against influenza, and 1 = had not received the vaccine against influenza. The response was coded as 0 if the parent had a medical contraindication for vaccination.

**Use of CAM.** To measure the use of CAM, the parents were presented with a list of 39 CAM items, from which they were asked to select the ones they had used during the past 12 months to treat an illness or to maintain good health. For the purpose of the present study, we included those CAM items that are not in the Finnish Current Care Guidelines [35], which are national evidence-based guidelines for the treatment and prevention of diseases in medical practice. The final list included the following 18 items: colloidal silver, turmeric, ginger, health powders, natural products for flu, aloe vera, kombucha, cupping, healing, laying on of hands, reiki, the Rosen method, zone therapy, salt therapy, chakra therapy, homeopathy, oil-pulling, and Ayurveda. The CAM variable was coded according to the number of CAM items used (0, 1, 2, 3, or 4 or more items).

**Reactance.** Trait reactance was measured with the 14-item version of the Hong Psychological Reactance Scale (HPRS; [36]). For each of the statements, the parents were asked to indicate their agreement on a scale from 1 (completely disagree) to 5 (completely agree). Only nine of those items were used in the analyses, based on a study that investigated the factor structure of the 14-item HPRS using the Finnish-speaking respondents of the present sample [37]. A higher HPRS score indicates higher trait reactance.

**Trust in doctors.** Six statements for measuring trust in doctors were created for the study (S1 Table; e.g., "I let doctors make the decisions concerning my health", "I trust doctors' ability to make correct diagnoses"). The statements were of varying polarity, and the participants were asked to indicate their agreement with each statement on a scale from 1 (completely disagree) to 4 (completely agree). Reverse-scored items were recoded so that a higher score indicated more trust.

**Vaccine attitudes.** Attitudes towards the benefit and safety of vaccines were measured with 15 statements created by the authors after literature review and discussions (S1 Table). The statements concerned childhood vaccines and vaccines in general (e.g., "The risk of side effects outweighs the protective benefits of childhood vaccines", "Vaccinating healthy children helps to protect others by stopping the spread of disease"), and influenza vaccines (e.g., "The risk of side effects outweighs the protective benefits of influenza vaccines", "It is not worth getting the influenza vaccine, as the influenza symptoms are not serious"). The participants were asked to indicate their agreement with each statement on a scale from 1 (completely disagree) to 4 (completely agree). The polarity of the statements varied, but the items were recoded so that higher scores indicated more positive attitudes.

## Statistical analyses

A preregistration of the statistical analyses can be found at [https://osf.io/wda4k?view_only=a3406aee4dbc45d2a094b56ec9a29525]. See S1 Preregistration for changes to the preregistered analyses. The analyses were conducted using structural equation modeling (SEM) in Mplus 8.4 [38]. SEM models can be used for modeling relationships between both latent and observed variables. As the present data collection was cross-sectional, the analyses cannot establish causality between the variables but allows us to test whether our data are consistent with a putative causal model. Trait reactance (Reactance; nine indicators), trust in doctors (Trust; six indicators), and vaccine attitudes (VaccAtt; 15 indicators) were represented by latent factors in the analyses. We first conducted confirmatory factor analyses (CFA) to test the fit of the factors.

 

Second, we assessed the zero-order correlations between all measures. Third, in an attempt to replicate the results of Hornsey et al. [17], who showed that higher trait reactance is related to more negative attitudes to vaccines, we examined whether reactance and trust in doctors predicted vaccine attitudes, by specifying a structural regression (SR) model with the vaccine attitudes factor as the outcome measure (Model 1). The vaccine attitudes factor was regressed on reactance and trust in doctors. Trust was also regressed on reactance to investigate whether trust mediated the associations between reactance and vaccine attitudes. Fourth, to examine our main research questions, we specified a similar SR model with vaccine behavior and CAM use as the outcome variables (Model 2). The outcome variables were again regressed on reactance and trust and trust was regressed on reactance.

As a fifth step, we investigated whether reactance and trust in doctors explained the possible associations between vaccination attitudes and CAM use, and vaccination behavior and CAM use. This was done by assessing whether the disturbance correlations between the outcome measures were weaker than the zero-order correlations between the outcomes. Disturbance correlations constitute the correlations between the proportions of the variances that are not explained by the model. If the disturbance correlations are weaker than the zero-order correlations, it means that the model explains variance that is shared between the outcome measures. To obtain the disturbance correlation between vaccine attitudes and CAM use, the CAM use measure was included as an outcome in Model 1.

Robust WLS (WLSMV) estimation was applied in the SR and CFA analyses, as the indicators and outcome variables were ordinal and responses were non-normally distributed. The relationships between the measures are represented by probit regression coefficients. This coefficient indicates the change in the outcome variable's standard normal distribution (z-score), given a one-unit increase in the predictor. As the data partly consisted of parents from the same family, responses can be considered clustered. Because of this, non-independence between observations was accounted for when computing standard errors and $\chi^2$ statistics in all analyses. Missing data were handled with pair-wise deletion.

## Results

The parents' responses on the outcome variables are presented in Table 2. A majority of the parents had never hesitated in a childhood vaccination decision or postponed or rejected a childhood vaccine. Half of the parents had taken the influenza vaccine the preceding season. Most parents reported that they had not used any of the CAM items during the past 12 months. The parents' responses to the statements of the four factors can be seen in S2 and S3 Tables.

### Latent factor modeling

The factors Reactance, $\chi^2(26) = 155.18$, CFI = .951, TLI = .932, RMSEA = .081; 90% CI[.069, .093], SRMR = .040, and Trust, $\chi^2(8) = 36.35$, CFI = .993, TLI = .987, RMSEA = .068; 90% CI [.047, .092], SRMR = .022, showed appropriate fit to the data with one correlated error term in each model. However, the fit of the factor VaccAtt was unsatisfactory, $\chi^2(90) = 816.57$, CFI = .858, TLI = .835, RMSEA = .103; 90% CI[.097, .110], SRMR = .076. The residual covariance matrix indicated that the model underestimated the relationships among the indicators concerning influenza vaccine attitudes, whereas the relationships between these indicators and the indicators measuring attitudes to childhood vaccines or vaccines in general, were overestimated. Modification indices also suggested the inclusion of several correlated error terms between the indicators measuring influenza vaccine attitudes. Therefore, we decided to split the VaccAtt factor into two factors: one with the indicators for attitudes towards childhood

**Table 2. Parents' responses concerning vaccination behavior and CAM use.**

| Variable | n | % |
|---|---|---|
| **Childhood vaccination** | | |
| No hesitation/postponing/rejection[a] | 559 | 73.46 |
| Hesitated | 187 | 24.57 |
| Postponed | 98 | 12.88 |
| Rejected | 55 | 7.22 |
| **Influenza vaccination** | | |
| Yes | 391 | 51.72 |
| No | 365 | 48.28 |
| **CAM use** | | |
| No | 483 | 62.73 |
| One item | 161 | 20.91 |
| Two items | 66 | 8.57 |
| Three items | 34 | 4.42 |
| Four or more items | 26 | 3.38 |

The responses to Hesitated, Postponed, and Rejected are not mutually exclusive, as a parent may have answered yes to all three questions.

[a]Includes nine individuals who reported that their child had medical contraindications for vaccination.

vaccines or vaccines in general (VaccAttGeneral), and one with the indicators for attitudes towards influenza vaccines (VaccAttFlu). Both VaccAttGeneral, $\chi^2(34) = 112.68$, CFI = .964, TLI = .953, RMSEA = .055; 90% CI[.044, .067], SRMR = .038, and VaccAttFlu, $\chi^2(4) = 31.60$, CFI = .992, TLI = .981, RMSEA = .095; 90% CI[.066, .127], SRMR = .019, fitted the data well after the inclusion of one correlated error term in each model. All residual correlations specified in the one-factor models were retained in the subsequent analyses.

The factor loadings and variances can be seen in S4 Table. Zero-order correlations between all measures are shown in Table 3. The relationship between reactance and trust was negative and statistically significant, indicating that individuals with higher trait reactance tended to have lower trust in doctors.

## Association between reactance and attitudes to vaccines

Due to the split of the vaccine attitudes factor, Model 1, investigating the relationship between reactance, trust, and vaccine attitudes, was re-specified to include two outcome measures: VaccAttGeneral and VaccAttFlu. The model showed good fit to the data, $\chi^2(395) = 862.25$, CFI = .955, TLI = .950, RMSEA = .039; 90% CI[.036, .043], SRMR = .052. The results revealed that reactance was directly and statistically significantly related to both VaccAttGeneral and VaccAttFlu (Table 4), indicating that parents with higher trait reactance had more negative attitudes to vaccines. Also, the indirect effects of reactance on both vaccine attitude measures, mediated by trust in doctors, were statistically significant. The total effect (the sum of direct and indirect effects) of reactance on VaccAttGeneral was $\beta = .27$, $SE = .04$, $t = 6.55$, $p < .001$, whereas the total effect on VaccAttFlu was $\beta = .25$, $SE = .04$, $t = 6.19$, $p < .001$.

## Association between reactance, vaccination behavior, and CAM use

The SR model including vaccination behavior and CAM use (Model 1) fitted the data well, $\chi^2(126) = 288.78$, CFI = .973, TLI = .967, RMSEA = .041; 90% CI[.035, .047], SRMR = .040. The model showed that reactance did not have a statistically significant direct effect on

**Table 3. Zero-order correlations between measures.**

| Measure | 1 | 2 | 3 | 4 | 5 | 6 | 7 |
|---|---|---|---|---|---|---|---|
| 1. Reactance | - | | | | | | |
| 2. Trust | -.33 | - | | | | | |
| 3. Childhood vaccination behavior | .16 | -.44 | - | | | | |
| 4. Influenza vaccination behavior | .18 | -.21 | .33 | - | | | |
| 5. CAM use | .08 | -.24 | .19 | .12 | - | | |
| 6. VaccAttGen | -.27 | .52 | -.66 | -.41 | -.24 | - | |
| 7. VaccAttFlu | -.25 | .48 | -.58 | -.78 | -.22 | .76 | - |

All other correlations statistically significant at $p < .001$, except for the correlations between reactance and childhood vaccination behavior ($p = .001$), and reactance and CAM use ($p = .087$). Reactance = trait reactance; Trust = trust in doctors; CAM = complementary and alternative medicine; VaccAttGen = attitudes towards childhood vaccines or vaccines in general; VaccAttFlu = attitudes towards influenza vaccines.

childhood vaccination behavior and CAM use ($\beta = .02$, $SE = .06$, $t = 0.35$, $p = .726$, and $\beta = .00$, $SE = .05$, $t = 0.02$, $p = .983$, respectively). We therefore compared a more parsimonious model, where these coefficients were constrained to zero, to the unconstrained model. The constrained model did not result in a statistically significant loss of fit, $\Delta\chi^2(2) = 0.16$, $p = .926$. Fig

**Table 4. Direct and indirect effects in the SR models.**

| | Path | | Unstandardized | | | | Standardized | | | |
|---|---|---|---|---|---|---|---|---|---|---|
| | | | $b$ | $SE$ | $t$ | $p$ | $\beta$ | $SE$ | $t$ | $p$ |
| **Model 1** | | | | | | | | | | |
| **Direct effects** | | | | | | | | | | |
| Reactance | | → Trust | -0.21 | 0.03 | 6.15 | < .001 | -.33 | .04 | 7.50 | < .001 |
| Reactance | | → VaccAttGeneral | -0.13 | 0.06 | 2.42 | .016 | -.12 | .05 | 2.39 | .017 |
| Reactance | | → VaccAttFlu | -0.15 | 0.06 | 2.30 | .021 | -.11 | .05 | 2.30 | .021 |
| Trust | | → VaccAttGeneral | 0.85 | 0.12 | 6.97 | < .001 | .48 | .04 | 11.30 | < .001 |
| Trust | | → VaccAttFlu | 0.95 | 0.12 | 7.70 | < .001 | .45 | .04 | 10.52 | < .001 |
| **Indirect effects** | | | | | | | | | | |
| Reactance | → Trust | → VaccAttGeneral | -0.18 | 0.04 | 4.99 | < .001 | -.16 | .03 | 5.69 | < .001 |
| Reactance | → Trust | → VaccAttFlu | -0.20 | 0.04 | 5.34 | < .001 | -.15 | .03 | 5.46 | < .001 |
| **Model 2** | | | | | | | | | | |
| **Direct effects** | | | | | | | | | | |
| Reactance | | → Trust | -0.20 | 0.03 | 6.18 | < .001 | -.33 | .04 | 7.53 | < .001 |
| Reactance | | → Influenza vaccine | 0.18 | 0.09 | 2.14 | .033 | .12 | .06 | 2.12 | .034 |
| Trust | | → Childhood vaccine | -1.13 | 0.15 | 7.75 | < .001 | -.45 | .05 | 9.83 | < .001 |
| Trust | | → Influenza vaccine | -0.44 | 0.15 | 2.98 | .003 | -.17 | .06 | 3.10 | .002 |
| Trust | | → CAM use | -0.62 | 0.13 | 4.68 | < .001 | -.24 | .05 | 5.23 | < .001 |
| **Indirect effects** | | | | | | | | | | |
| Reactance | → Trust | → Childhood vaccine | 0.23 | 0.04 | 5.49 | < .001 | .15 | .03 | 5.55 | < .001 |
| Reactance | → Trust | → Influenza vaccine | 0.09 | 0.03 | 2.73 | .006 | .06 | .02 | 2.72 | .006 |
| Reactance | → Trust | → CAM use | 0.12 | 0.03 | 4.04 | < .001 | .08 | .02 | 4.00 | < .001 |

Reactance = trait reactance; Trust = trust in doctors; Influenza vaccine = influenza vaccination behavior; Childhood vaccine = childhood vaccination behavior; CAM = complementary and alternative medicine; VaccAttGen = attitudes towards childhood vaccines or vaccines in general; VaccAttFlu = attitudes towards influenza vaccines.

1 displays the final model, $\chi^2(128) = 276.12$, CFI = .975, TLI = .970, RMSEA = .039; 90% CI [.032, .045], SRMR = .040.

Reactance had a small and statistically significant direct effect on influenza vaccination behavior (Table 4), indicating that parents with higher trait reactance were less likely to have taken the influenza vaccine during the previous influenza season. Furthermore, reactance had small and statistically significant indirect effects on all outcome measures that were mediated by trust in doctors. Hence, the results were consistent with a model where individuals with higher trait reactance are more likely to have lower trust in doctors, and as a consequence, are more likely to have rejected a childhood vaccine for their children and the influenza vaccine for themselves, and to use more CAM. The total effect of reactance on influenza vaccination was $\beta = .17$, $SE = .05$, $t = 3.44$, $p = .001$.

## Zero-order and disturbance correlations

Tables 5 and 6 show the zero-order correlations between the measures for vaccination behavior, vaccine attitudes, and CAM use, as well as their disturbance correlations from Model 1 with CAM use included, $\chi^2(421) = 887.86$, CFI = .956, TLI = .951, RMSEA = .038; 90% CI [.034, .041], SRMR = .051), and from the un-constrained Model 2. When it comes to the association between vaccination behavior and CAM use, the disturbance correlations between the outcome variables, after controlling for Trust and Reactance, were lower than the zero-order correlations. However, the confidence intervals of the disturbance correlations were wide and overlapped with the zero-order correlations, suggesting that trait reactance and trust in doctors do not explain the association between vaccination behavior and CAM use. The difference between zero-order correlations and disturbance correlations was larger for the association between vaccine attitudes and CAM than for vaccination behavior and CAM, and the confidence intervals for the disturbance correlations showed minimal overlap with the zero-order correlations. This suggests that reactance and trust may explain a small part of the association between vaccine attitudes and CAM use. It is, however, important to note that the zero-order correlations between the vaccination-related variables and the CAM use variable were small ($r$ range: .12-.24).

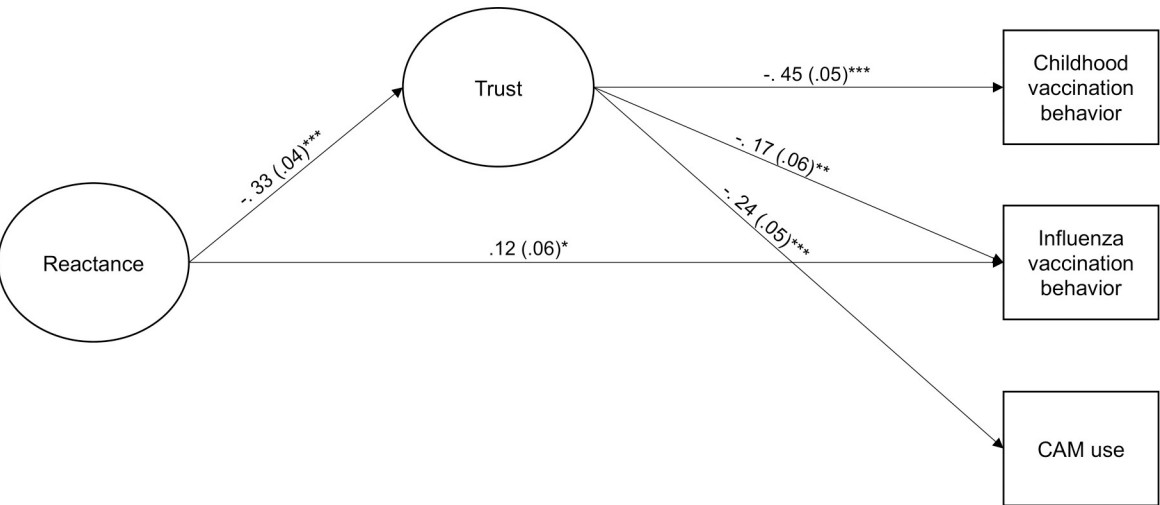

**Fig 1. Standardized estimates (standard errors) from model 2.** Factor indicators, loadings, and variances, as well as disturbances and their covariances are not shown in the figure. The paths from Reactance to Childhood vaccine and CAM use are set to zero. * $p < .05$; ** $p < .01$; *** $p < .001$.

**Table 5. Zero-order and disturbance correlations [95% CI] between childhood vaccination behavior, influenza vaccination behavior, and CAM use.**

| Outcome variable | Zero-order | | | Disturbance | | |
|---|---|---|---|---|---|---|
| | 1 | 2 | 3 | 1 | 2 | 3 |
| 1. Childhood vaccination behavior | - | | | - | | |
| 2. Influenza vaccination behavior | .33*** [.21, .44] | - | | .26*** [.14, .39] | - | |
| 3. CAM use | .19*** [.09, .29] | .12* [.01, .22] | - | .10 [-.01, .21] | .07 [-.04, .18] | - |

## Discussion

Previous studies have suggested that an unwillingness to agree with the medical consensus may lie behind both negative attitudes to vaccines and positive attitudes to CAM [27,28,32]. The present study investigated this idea further by exploring if trait reactance plays a part in vaccine attitudes, vaccination decisions, and in the use of CAM, and whether these relationships are mediated by trust in doctors. To the best of our knowledge, this is the first study that examined the association between trait reactance and actual vaccination behavior, and that jointly investigated the role of trait reactance in predicting vaccine attitudes and CAM use, and vaccination behavior and CAM use.

The results from the present study, conducted in a relatively large sample ($N = 770$) of Finnish parents of young children, showed that trait reactance had a statistically significant direct effects on the parents' attitudes to influenza vaccines and to vaccines in general, indicating that higher trait reactance was related to more negative attitudes to vaccines. These results are in line with the study by Hornsey et al. [17]. Our findings, however, shed more light on the relationship between trait reactance and vaccine attitudes by specifying that trust in doctors plays an important role in the association.

Concerning actual vaccination behavior, trait reactance had a small direct effect on the parents' decision to take the influenza vaccine, but there was no direct effect of trait reactance on the parents' decisions to accept childhood vaccinations for their children. However, as was the case with vaccine attitudes, all indirect paths between trait reactance and the vaccination behavior variables were statistically significant, meaning that parents with higher trait reactance had less trust in doctors and a smaller likelihood of having accepted vaccines for their children and for themselves. Our results thus extend previous research by showing that trait reactance not only affects attitudes to vaccines, but it has small effects on the actual vaccination decisions as well. The finding that parents with more trust in doctors were more likely to have accepted vaccinations, and more likely to have positive attitudes to vaccines, was also in line with previous studies (for reviews, see e.g., [5,9–15]).

Based on the results, it seems that trait reactance and trust in doctors explain somewhat more of the variance in vaccine attitudes than in actual vaccination behavior. Also, the relationship between vaccination attitudes and CAM use is slightly stronger than the one between vaccination behavior and CAM use. One possible explanation for this discrepancy is that embracing anti-vaccination attitudes may be a way of expressing one's personal identity and of

**Table 6. Zero-order and disturbance correlations [95% CI] between VaccAttGen, VaccAttFlu, and CAM use.**

| Outcome variable | Zero-order | | | Disturbance | | |
|---|---|---|---|---|---|---|
| | 1 | 2 | 3 | 1 | 2 | 3 |
| 1. VaccAttGen | - | | | - | | |
| 2. VaccAttFlu | .76 [.72, .81] | - | | .68 [.62, .74] | - | |
| 3. CAM use | -.24 [-.33, -.16] | -.22 [-.31, -.13] | - | -.14 [-.25, -.04] | -.12 [-.22, -.02] | - |

communicating that to others [39]. However, when it comes down to the actual vaccination decision, it is possible that also people who express anti-vaccination attitudes choose vaccinations after all.

Trait reactance did not have a direct effect on parents' CAM use, but in the same way as for vaccination attitudes and behavior, there was a statistically significant indirect effect of trait reactance that went via trust in doctors. As expected based on previous studies [27,28,32], the effects of the predictors on CAM use were in the opposite direction, compared to their effects on vaccination behavior, as higher trait reactance was associated with less trust in doctors, which in turn was associated with more use of CAM treatments and substances that are not included in evidence-based medicine.

Taken together, the results of the present study are thus consistent with a model that suggests that one of the reasons why some individuals high in trait reactance have negative attitudes to vaccines, do not accept vaccines for their children and for themselves, and use CAM, is that they have low trust in doctors.

We also tested the speculations put forth in previous studies [27,28,32] that negative attitudes to vaccines and positive attitudes to CAM may be driven by a shared underlying reluctance to agree with the medical consensus. Our analyses indeed revealed weak associations between vaccine attitudes and CAM use ($r = -.22 – -.24$), which were roughly in line with previous research [31,32], and between vaccination behavior and CAM use ($r = .12 – .19$), but there was no clear support for the hypothesis that the associations would be explained by trait reactance and trust in doctors. These findings suggest that high trait reactance and low trust in doctors has different consequences for different people. In some individuals, high trait reactance and a distrust in doctors might result in anti-vaccination attitudes and behavior, while for others, they might lead to CAM use. One possible explanation for this is that people vary in what is important to them, leading them to react against different aspects of conventional medicine.

## Limitations

As the present study employs a cross-sectional design, all causal interpretations are speculative. However, trait reactance refers to the predisposition to act with reactance in situations that are perceived as threats to the freedom of choice [22]. Individuals who tend to be reactant may embrace attitudes or engage in behavior that go against the option that has been imposed on them. Therefore, the present study assumes that trait reactance results in attitudes and behavior (i.e., distrust in medical doctors, anti-vaccination attitudes and behavior, and use of CAM), and not the other way around.

Another limitation that may affect the validity of the results, is the fact that the present study is based on self-reported attitudes and behavior. The responses may thus have been influenced by factors such as social desirability bias or memory issues. Also, the questionnaires regarding vaccine attitudes, trust in doctors, and CAM use, have not been validated in other samples. However, during the process of developing the questionnaires for the present study, experts in the field assessed the face validity of the questions. Also, when it comes to the questionnaires probing vaccine attitudes and trust in doctors, factor analysis was used to assess the factor loadings of the questions on the constructs and to handle measurement error.

Concerning possible limitations to generalizability, the parents in the present study are part of a birth cohort study that includes health-related measurements during multiple time points over several years [34]. It is therefore possible that these parents have higher trust in doctors and are less reactant than the general population. Finally, the response-rate was rather low, which possibly resulted in selection bias.

## Conclusions

The results from the present study involving Finnish parents of young children show that parents with higher trait reactance are more likely to distrust doctors, and because of that, have more negative attitudes to vaccines, and have a higher likelihood of not accepting vaccines for their children and themselves. Parents with higher trait reactance and a distrust in doctors are also more likely to turn to CAM treatments and substances that are not included in evidence-based medicine. Furthermore, high trait reactance and low trust in doctors have different consequences for different people. In some individuals, high trait reactance and a distrust in doctors might result in anti-vaccination attitudes and behavior, while in others, they might lead to CAM use. One possible explanation for this is that people vary in what is important to them, leading them to react against different aspects of conventional medicine.

However, even though reactance is important to keep in mind when addressing parents' concerns about vaccines, it is important to note that the parents' use of CAM, their attitudes towards vaccines, and their decisions to accept or reject vaccines, are mainly due to other factors than trait reactance. Also, parents with high trait reactance constitute a clear minority and it would therefore seem plausible to assume that the main focus when trying to increase immunization rates should still be on the non-reactant parents. This is also supported by the results from a recent study showing that mandatory vaccinations are associated with higher vaccination coverage [40].

## Supporting information

**S1 Table. Survey questions measuring trust in doctors and vaccine attitudes.**
(DOCX)

**S2 Table. Parents' responses to the included items of the HPRS.**
(DOCX)

**S3 Table. Parents' responses to statements measuring trust in doctors and attitudes to vaccines.**
(DOCX)

**S4 Table. Factor loadings and variances from confirmatory factor analyses.**
(DOCX)

**S1 Preregistration. Transparent changes.**
(DOCX)

**S1 Questionnaire. Questionnaire translated into English.**
(DOCX)

**S2 Questionnaire. Questionnaire in Swedish.**
(DOCX)

**S3 Questionnaire. Questionnaire in Finnish.**
(DOCX)

## Author Contributions

**Conceptualization:** Anna Soveri, Linda C. Karlsson, Jan Antfolk, Stephan Lewandowsky.

**Formal analysis:** Anna Soveri, Linda C. Karlsson, Jan Antfolk, Otto Waris.

**Funding acquisition:** Anna Soveri, Hasse Karlsson, Linnea Karlsson, Mikael Lindfelt.

**Investigation:** Anna Soveri, Linda C. Karlsson, Otto Mäki.

**Methodology:** Anna Soveri, Linda C. Karlsson, Jan Antfolk, Hasse Karlsson, Linnea Karlsson, Stephan Lewandowsky.

**Project administration:** Anna Soveri, Mikael Lindfelt.

**Resources:** Anna Soveri, Hasse Karlsson, Linnea Karlsson.

**Supervision:** Anna Soveri, Jan Antfolk, Stephan Lewandowsky.

**Writing – original draft:** Anna Soveri, Linda C. Karlsson.

**Writing – review & editing:** Anna Soveri, Linda C. Karlsson, Otto Mäki, Jan Antfolk, Otto Waris, Hasse Karlsson, Linnea Karlsson, Mikael Lindfelt, Stephan Lewandowsky.

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
