## [Decision Letter · Decision Letter 0]

7 May 2020

PONE-D-20-10006

Trait Reactance as a Predictor of Vaccination Behavior and Use of Complementary and Alternative Medicine in Parents of Young Children

PLOS ONE

Dear Mrs Soveri,

Thank you for submitting your manuscript to PLOS ONE. After careful consideration, we feel that it has merit but does not fully meet PLOS ONE’s publication criteria as it currently stands. Therefore, we invite you to submit a revised version of the manuscript that addresses the points raised during the review process.

We would appreciate receiving your revised manuscript by Jun 21 2020 11:59PM. To enhance the reproducibility of your results, we recommend that if applicable you deposit your laboratory protocols in protocols.io, where a protocol can be assigned its own identifier (DOI) such that it can be cited independently in the future. For instructions see: http://journals.plos.org/plosone/s/submission-guidelines#loc-laboratory-protocols

We look forward to receiving your revised manuscript.

Kind regards,

Peter Karl Jonason

Academic Editor

PLOS ONE

Journal Requirements:

Reviewers' comments:

Reviewer's Responses to Questions

**Comments to the Author**

1. Is the manuscript technically sound, and do the data support the conclusions?

Reviewer #1: Yes

Reviewer #2: Partly

2. Has the statistical analysis been performed appropriately and rigorously? 

Reviewer #1: Yes

Reviewer #2: No

3. Have the authors made all data underlying the findings in their manuscript fully available?

Reviewer #1: Yes

Reviewer #2: Yes

4. Is the manuscript presented in an intelligible fashion and written in standard English?

Reviewer #1: Yes

Reviewer #2: Yes

5. Review Comments to the Author

Reviewer #1: I enjoyed reading this article and think it would be a good fit for PLoSONE. The writing is clear, the lit review thorough, and the data set is impressive. Analyses seem appropriate, and limitations are laid out in a clear and non-defensive fashion. I know that novelty is not necessarily a primary criterion for acceptance in PLoSONE, but I note that this is the first study that I am aware of that has explored relationships between reactance and vaccination behavior (as opposed to attitudes).

Some minor suggestions for improvement:

• I’d prefer if you expanded the correlation table to incorporate the outcome variables as well. I always find correlations reassuring, particularly when there’s the hint of suppression

• I also like to see authors provide a little insight into measures in the main manuscript, rather than just referring people to supplementary files. I’m thinking of the attitudes in particular: 2 or 3 example items would be fine.

• I will send you directly some recent papers of mine that might be relevant to the current analysis: one in JESP and one in SS&M (both are currently in press but can be found online I think). The paper in SS&M is particularly relevant, as it examines the relationship between CAM use, vaxx attitudes, and trust in various medical interventions. The message of the paper is that anti-vaxx attitudes are associated with mistrust of conventional medicine … not so much trust in CAM. So it’s a suspicion issue primarily: people are turning away from vaccinations because they don’t trust conventional medicine, and they are turning to CAM for the same reason. This parallels your data: most of the effect between trait reactance and CAM use is mediated by trust in doctors. It suggest that this mistrust issue is the proximal cause of both antivaxx attitudes and CAM use (and that the link between CAM use and anti-vaxx attitudes might be partly an artifact of having this one predictor in common)

• I found it interesting that reactance predicted attitudes (as in Hornsey et al., 2018) but not behaviours. I think you could make more of that in the Discussion. It suggests to me that for people high in reactance, anti-vaxx views serve a performative, identity-expressive function (communicating something about who you are). But when it comes down to it, these people also go for the safety of the vaccination. In other words they engage in cheap antivaxx talk but they take the vaccinations anyway. Interesting! I talk a bit about this in the Hornsey & Fielding American Psych paper on personal identity expression, and your data provide a good circumstantial case for this process.

That’s it! I thank you for the chance to read your paper, and wish you all the best with your ongoing program of research.

Matthew Hornsey

Reviewer #2: Summary

This is an interesting article that furthers the investigation of the psychology of anti-vaccination attitudes and scepticism towards conventional medicine more generally.

The writing is generally good, and the literature reviewed is comprehensive and pertinent. However, as discussed below, the MS still requires more proof reading to correct a number of minor expression issues throughout.

My version appeared to be missing Figure labels, which contributed to ambiguity in understanding the results. For example, I am assuming the reported effects are standardised, but I'm not sure.

Although the analysis and conclusions are reasonable, I strongly advise that the authors consider a simpler analysis technique. The CFA is sound, but the causal effects can be more transparently reported using conventional regressions. This is related to my concerns regarding the causal structure, and also my feeling that investigating the mediating role of attitudes on behaviour is not very interesting, and tangential to the main focus of the paper.

The overly-complex analysis leads to a lack of confidence on the part of the reader, which is a shame given the hypotheses are simple, and the variables are few. To illustrate, one result the reader will be interested in is comparing the relative influence of Trust and Reactance on CAM and VAX. It appears that Trust might be more important, but even this elementary result is somewhat obscured.

Overall, I believe the article is fundamentally sound, and should ultimately be published. With simplification of the analyses so that the results are more transparent, and a basic edit, I believe it could be.

Details

One question I have in relation to the rationale for the research question provided in the introduction. The reasoning for trait reactance being an explanatory factor in vaccine scepticism is set out well. However, I don't see a similar argument for CAM? Perhaps the idea is that CAM adherents are motivated by reactance against conventional medical advice more generally - but this seems a little more tenuous as compared to vaccination attitudes. Can you please address this issue?

A related point is that the rationale for trust in doctors mediating the effect of reactance on CAM and anti-vac. It's not clear to me that it should be a mediating effect. Indeed, it seems at least (or more) plausible that trust in doctors would add to, or exacerbate (captured by an interaction) the direct effect of reactance. When adopting a path analytic or SEM approach, it's very important to have a very strong rationale for the proposed model being much more plausible than alternative formulations. Can this be provided? Alternatively, the plausible models might be fit and compared.

278 - small correction to terminology. Since you're using SEM, your three constructs are not technically measured, they're latent

Figures - the labelling needs to be improved, and formatted so they are contained within the boxes. 'CV' and 'IV' for example, would benefit from more informative labels. I think the structure was created using automatic software. Manual formatting using software (I can recommend OmniGraffle) is necessary. These diagrams also usually include * and/r standard errors.

Frankly, after seeing the SEM structure, in which (almost) everything is related to everything else, I am more uncomfortable about this analytic approach. SEM or PA models are essentially defined by the causal links that are *not* in the model. We usually are motivated to apply SEM/PA when we hypothesis a much simpler structure than the correlation matrix. Further, especially in an exploratory context, SEM/PA is focused on comparing alternative plausible models.

I think the CFA approach used to refine the constructs (e.g. vaccination attitudes) makes sense. However, the subsequent analyses become very complicated, for an analysis that involves just a few variables. I'm also struggling to relate the beta coefficients mentioned in the text and diagram to those reported in Table 4. None of the standardised effects in Table 4 exceed .21, yet mention of direct, indirect and total effects in the text are often greater.

Overall, the large number of 'effects' the reader has to wade through tends to obscure the results. The issue is compounded by the issue mentioned earlier, whereby specifying that reactances causes (decrease in) trust.

Both those issues could be resolved by putting aside that causal assumption, and specifying simply:

1. Reactance and Trust causes (with potential interaction) vaccination attitudes

2. Trust and reactance cause CV, IV, CAM

There's no real benefit to including attitudes as mediating variables in the main model, since it's trivially true that attitudes drive behaviour. It greatly complicates the results, without providing any real benefit.

This could be done with some regressions. They will allow you, for instance, to provide a straight-forward comparison of the relative influence on trait reactance on anti-vax and CAM.

The reader will have much more confidence in the results, and they will be much more transparent, if the analyses can be simplified. Less is definitely more, when it comes to statistical analyses.

It is beyond the scope of the review process to provide detailed close editing when there are a great many required edits. I have identified expression issues in the first page. However, the authors will need to take steps to improve expression throughout the manuscript. The issues are generally quite minor, but polished expression is needed for journal publication.

64 - widely regarded

70 - Salmon et al

71 delete great

72 can lead to the

76 delete great

76 the decision to vaccinate

80 vaccination decision-making

81 delete for instance

83 "actors in the vaccine chain" odd wording

6. PLOS authors have the option to publish the peer review history of their article (what does this mean?). If published, this will include your full peer review and any attached files.

Reviewer #1: No

Reviewer #2: No

---

## [Author Response · Author response to Decision Letter 0]

23 Jun 2020

Response to reviewers

Editor’s comments

Journal Requirements:

Response: The manuscript has now been formatted according to PLOS ONE’s style requirements.

Response: The parts of the questionnaire that have been developed for this study are now included as supporting information in English, Swedish, and Finnish (S1-S3 Questionnaires).

Response: Due to Finnish federal legislation on personal data protection in medical research, the original research data cannot be made available online, but data can potentially be shared with Material Transfer Agreement. Requests and collaboration initiatives can be directed to the Board of the FinnBrain Birth Cohort Study. Please contact data manager Teemu Kemppainen (teekem@utu.fi).

Response: Captions and in-text citations have been formatted according to PLOS ONE’s requirements.

Reviewer #1

I enjoyed reading this article and think it would be a good fit for PLoSONE. The writing is clear, the lit review thorough, and the data set is impressive. Analyses seem appropriate, and limitations are laid out in a clear and non-defensive fashion. I know that novelty is not necessarily a primary criterion for acceptance in PLoSONE, but I note that this is the first study that I am aware of that has explored relationships between reactance and vaccination behavior (as opposed to attitudes).

Response: We thank the reviewer for the positive feedback. We now highlight the novelty of the study both in the Introduction and the Discussion:

“To the best of our knowledge, this is the first study to look at actual vaccination behavior in this context.” (p. 7)

“To the best of our knowledge, this is the first study that examined the association between trait reactance and actual vaccination behavior, and that jointly investigated the role of trait reactance in predicting vaccine attitudes and CAM use, and vaccination behavior and CAM use.”(p. 20) 

Some minor suggestions for improvement:

1. I’d prefer if you expanded the correlation table to incorporate the outcome variables as well. I always find correlations reassuring, particularly when there’s the hint of suppression

Response: The table (Table 3) now includes the zero-order correlations between all measures.

2. I also like to see authors provide a little insight into measures in the main manuscript, rather than just referring people to supplementary files. I’m thinking of the attitudes in particular: 2 or 3 example items would be fine.

Response: Based on the Reviewer’s suggestion, we have added example statements for the Trust and Attitudes measures in the Method section.

3. I will send you directly some recent papers of mine that might be relevant to the current analysis: one in JESP and one in SS&M (both are currently in press but can be found online I think). The paper in SS&M is particularly relevant, as it examines the relationship between CAM use, vaxx attitudes, and trust in various medical interventions. The message of the paper is that anti-vaxx attitudes are associated with mistrust of conventional medicine … not so much trust in CAM. So it’s a suspicion issue primarily: people are turning away from vaccinations because they don’t trust conventional medicine, and they are turning to CAM for the same reason. This parallels your data: most of the effect between trait reactance and CAM use is mediated by trust in doctors. It suggest that this mistrust issue is the proximal cause of both antivaxx attitudes and CAM use (and that the link between CAM use and anti-vaxx attitudes might be partly an artifact of having this one predictor in common)

Response: We thank the reviewer for sending us the relevant recent papers. We have added them to the manuscript. 

4. I found it interesting that reactance predicted attitudes (as in Hornsey et al., 2018) but not behaviours. I think you could make more of that in the Discussion. It suggests to me that for people high in reactance, anti-vaxx views serve a performative, identity-expressive function (communicating something about who you are). But when it comes down to it, these people also go for the safety of the vaccination. In other words they engage in cheap antivaxx talk but they take the vaccinations anyway. Interesting! I talk a bit about this in the Hornsey & Fielding American Psych paper on personal identity expression, and your data provide a good circumstantial case for this process.

Response: We wish to thank the Reviewer for the interesting idea. We have added the following section to the Discussion:

“Based on the results, it seems that trait reactance and trust in doctors explain somewhat more of the variance in vaccine attitudes than in actual vaccination behavior. Also, the relationship between vaccination attitudes and CAM use is slightly stronger than the one between vaccination behavior and CAM use. One possible explanation for this discrepancy is that embracing anti-vaccination attitudes may be a way of expressing one’s personal identity and of communicating that to others (39). However, when it comes down to the actual vaccination decision, it is possible that also people who express anti-vaccination attitudes choose vaccinations after all.” (pp. 21-22)

That’s it! I thank you for the chance to read your paper, and wish you all the best with your ongoing program of research.

Matthew Hornsey

Reviewer #2 

Summary

This is an interesting article that furthers the investigation of the psychology of anti-vaccination attitudes and scepticism towards conventional medicine more generally. The writing is generally good, and the literature reviewed is comprehensive and pertinent. However, as discussed below, the MS still requires more proof reading to correct a number of minor expression issues throughout.

Response: We thank the Reviewer for the constructive feedback. Please see the responses to the specific comments below.

1. My version appeared to be missing Figure labels, which contributed to ambiguity in understanding the results. For example, I am assuming the reported effects are standardised, but I'm not sure.

Response: In line with PLOS ONE instructions for figures, the Figure captions and notes can be found in the manuscript immediately following the paragraph where the figure is first cited (e.g., Figure 1 on page 18), and not together with the figure. 

The effects reported in Fig 1 are indeed standardized. This information has now been added to the figure legend: “Fig 1. Standardized Estimates (standard errors) from Model 2”(p.18)

2. Although the analysis and conclusions are reasonable, I strongly advise that the authors consider a simpler analysis technique. The CFA is sound, but the causal effects can be more transparently reported using conventional regressions. This is related to my concerns regarding the causal structure, and also my feeling that investigating the mediating role of attitudes on behaviour is not very interesting, and tangential to the main focus of the paper.

The overly-complex analysis leads to a lack of confidence on the part of the reader, which is a shame given the hypotheses are simple, and the variables are few. To illustrate, one result the reader will be interested in is comparing the relative influence of Trust and Reactance on CAM and VAX. It appears that Trust might be more important, but even this elementary result is somewhat obscured.

Overall, I believe the article is fundamentally sound, and should ultimately be published. With simplification of the analyses so that the results are more transparent, and a basic edit, I believe it could be.

Response: We have simplified the analyses (described in more detail in our response to Reviewer 2, comment 7). We have kept the analyses within a SEM environment to be able to include the latent constructs in the analyses. Both the relative influence of Trust and Reactance on the outcome measures, as well as the theorized mediating effect of Trust, are more easily interpreted from the new models. 

Details

3. One question I have in relation to the rationale for the research question provided in the introduction. The reasoning for trait reactance being an explanatory factor in vaccine scepticism is set out well. However, I don't see a similar argument for CAM? Perhaps the idea is that CAM adherents are motivated by reactance against conventional medical advice more generally - but this seems a little more tenuous as compared to vaccination attitudes. Can you please address this issue?

Response: We wish to thank the reviewer for pointing out that the justification for the hypothesis that trait reactance would predict CAM, was missing in the manuscript. We apologize for this and have added the following paragraph to the Introduction: “The assumption that trait reactance plays a role also in the decision to use CAM, is based on the idea that CAM represents nonconventional treatments that fall outside the prevailing medical recommendations. Reactance may thus manifest itself in use of CAM in individuals who experience conventional medicine as a threat to their freedom of choice.” (p.7)

4. A related point is that the rationale for trust in doctors mediating the effect of reactance on CAM and anti-vac. It's not clear to me that it should be a mediating effect. Indeed, it seems at least (or more) plausible that trust in doctors would add to, or exacerbate (captured by an interaction) the direct effect of reactance. When adopting a path analytic or SEM approach, it's very important to have a very strong rationale for the proposed model being much more plausible than alternative formulations. Can this be provided? Alternatively, the plausible models might be fit and compared.

Response: The rationale behind the mediation model is that we measure trait reactance and not state reactance. Our theory thus suggests that trait reactant individuals will, as a consequence of their reactance, also distrust medical authorities such as doctors, and that this increased distrust manifests itself also as negative attitudes to vaccines. We discuss this in the revised Introduction and Limitation sections: 

“Reactance may manifest itself as negative attitudes towards vaccines and medical authorities, and in some individuals, even in a behavior that favors the option that they feel has been taken away from them, that is, to postpone vaccinations or to not get vaccinated altogether.” (p. 5)

and

 “As the present study employs a cross-sectional design, all causal interpretations are speculative. However, trait reactance refers to the predisposition to act with reactance in situations that are perceived as threats to the freedom of choice (22). Individuals who tend to be reactant may embrace attitudes or engage in behavior that go against the option that has been imposed on them. Therefore, the present study assumes that trait reactance results in attitudes and behavior (i.e., distrust in medical doctors, anti-vaccination attitudes and behavior, and use of CAM), and not the other way around.”(p.23) 

5. 278 - small correction to terminology. Since you're using SEM, your three constructs are not technically measured, they're latent

Response: Based on the Reviewer’s comment, we have checked the manuscript for inaccuracies in terminology. The text on row 278 in the first version of the manuscript referred to the observed outcome measures, which are not latent.

6. Figures - the labelling needs to be improved, and formatted so they are contained within the boxes. 'CV' and 'IV' for example, would benefit from more informative labels. I think the structure was created using automatic software. Manual formatting using software (I can recommend OmniGraffle) is necessary. These diagrams also usually include * and/r standard errors.

Response: We apologize for using unclear labels in the figure and have corrected this in the revised manuscript. We have formatted the diagram and include standard errors and significance levels.

7. Frankly, after seeing the SEM structure, in which (almost) everything is related to everything else, I am more uncomfortable about this analytic approach. SEM or PA models are essentially defined by the causal links that are *not* in the model. We usually are motivated to apply SEM/PA when we hypothesis a much simpler structure than the correlation matrix. Further, especially in an exploratory context, SEM/PA is focused on comparing alternative plausible models.

I think the CFA approach used to refine the constructs (e.g. vaccination attitudes) makes sense. However, the subsequent analyses become very complicated, for an analysis that involves just a few variables. I'm also struggling to relate the beta coefficients mentioned in the text and diagram to those reported in Table 4. None of the standardised effects in Table 4 exceed .21, yet mention of direct, indirect and total effects in the text are often greater.

Overall, the large number of 'effects' the reader has to wade through tends to obscure the results. The issue is compounded by the issue mentioned earlier, whereby specifying that reactances causes (decrease in) trust.

Both those issues could be resolved by putting aside that causal assumption, and specifying simply:

1. Reactance and Trust causes (with potential interaction) vaccination attitudes

2. Trust and reactance cause CV, IV, CAM

There's no real benefit to including attitudes as mediating variables in the main model, since it's trivially true that attitudes drive behaviour. It greatly complicates the results, without providing any real benefit.

This could be done with some regressions. They will allow you, for instance, to provide a straight-forward comparison of the relative influence on trait reactance on anti-vax and CAM.

The reader will have much more confidence in the results, and they will be much more transparent, if the analyses can be simplified. Less is definitely more, when it comes to statistical analyses.

Response: We have simplified the analyses according to the Reviewer’s suggestions, with one exception. Instead of the suggested interaction between Reactance and Trust, we still include Trust as a mediator, as this is central for our hypotheses (please see our response to Reviewer 2, comment 4). We conduct two structural regression models: 1) one where Reactance and Trust predict vaccination behavior and CAM use, and 2) another one where Reactance and Trust predict vaccine attitudes. In both models, Trust is also regressed on Reactance to investigate the mediation hypothesis. This change addresses the Reviewer’s concern on whether the SEM approach is motivated, as all paths in the models now specifically test our hypotheses. The mediating role of attitudes on behavior, for which we did not specify a hypothesis, has been omitted. The new analysis scheme also considerably simplifies the interpretation of the results. To further utilize the SEM approach, we re-specify paths that show very weak and non-significant associations to be fixed to zero, and test whether the fit is significantly reduced. 

We thank the Reviewer for the suggestions and agree that the simplified analysis scheme is easier to interpret.

8. It is beyond the scope of the review process to provide detailed close editing when there are a great many required edits. I have identified expression issues in the first page. However, the authors will need to take steps to improve expression throughout the manuscript. The issues are generally quite minor, but polished expression is needed for journal publication.

64 - widely regarded

70 - Salmon et al

71 delete great

72 can lead to the

76 delete great

76 the decision to vaccinate

80 vaccination decision-making

81 delete for instance

83 "actors in the vaccine chain" odd wording

Response: We have corrected all language issues pointed out by the Reviewer. We have also polished the language throughout the manuscript and changed the reference style to the Vancouver system.

---

## [Decision Letter · Decision Letter 1]

4 Jul 2020

PONE-D-20-10006R1

Trait reactance and trust in doctors as predictors of vaccination behavior, vaccine attitudes, and use of complementary and alternative medicine in parents of young children

PLOS ONE

Dear Dr. Soveri,

Thank you for submitting your manuscript to PLOS ONE. After careful consideration, we feel that it has merit but does not fully meet PLOS ONE’s publication criteria as it currently stands. Therefore, we invite you to submit a revised version of the manuscript that addresses the points raised during the review process.

We look forward to receiving your revised manuscript.

Kind regards,

Peter Karl Jonason

Academic Editor

PLOS ONE

Reviewers' comments:

Reviewer's Responses to Questions

**Comments to the Author**

1. If the authors have adequately addressed your comments raised in a previous round of review and you feel that this manuscript is now acceptable for publication, you may indicate that here to bypass the “Comments to the Author” section, enter your conflict of interest statement in the “Confidential to Editor” section, and submit your "Accept" recommendation.

Reviewer #1: All comments have been addressed

Reviewer #2: All comments have been addressed

2. Is the manuscript technically sound, and do the data support the conclusions?

Reviewer #1: Yes

Reviewer #2: Yes

3. Has the statistical analysis been performed appropriately and rigorously? 

Reviewer #1: Yes

Reviewer #2: Yes

4. Have the authors made all data underlying the findings in their manuscript fully available?

Reviewer #1: Yes

Reviewer #2: Yes

5. Is the manuscript presented in an intelligible fashion and written in standard English?

Reviewer #1: Yes

Reviewer #2: Yes

6. Review Comments to the Author

Reviewer #1: The authors have taken care in the revision and have addressed all my original concerns. My only remaining comment: in revising the abstract, the conclusions section contains quite a bit of causal language, which should be avoided given that it's a correlational study. Apart from this I have no further comments and I don't need to see a revision. I look forward to seeing the paper in print at some stage soon.

All the best, Matthew

Reviewer #2: Thanks for addressing my comments seriously. As well as being more polished overall, the revised analyses and results provide a much clearer and less ambiguous interpretation, which well supports the conclusions. An interesting result, and good work.

7. PLOS authors have the option to publish the peer review history of their article (what does this mean?). If published, this will include your full peer review and any attached files.

Reviewer #1: **Yes: **Matthew Hornsey

Reviewer #2: No

---

## [Author Response · Author response to Decision Letter 1]

5 Jul 2020

PONE-D-20-10006R1

Trait reactance and trust in doctors as predictors of vaccination behavior, vaccine attitudes, and use of complementary and alternative medicine in parents of young children

PLOS ONE

Dear Dr. Soveri,

Thank you for submitting your manuscript to PLOS ONE. After careful consideration, we feel that it has merit but does not fully meet PLOS ONE’s publication criteria as it currently stands. Therefore, we invite you to submit a revised version of the manuscript that addresses the points raised during the review process.

We look forward to receiving your revised manuscript.

Kind regards,

Peter Karl Jonason

Academic Editor

PLOS ONE

Reviewers' comments:

Reviewer's Responses to Questions

Comments to the Author

1. If the authors have adequately addressed your comments raised in a previous round of review and you feel that this manuscript is now acceptable for publication, you may indicate that here to bypass the “Comments to the Author” section, enter your conflict of interest statement in the “Confidential to Editor” section, and submit your "Accept" recommendation.

Reviewer #1: All comments have been addressed

Reviewer #2: All comments have been addressed

2. Is the manuscript technically sound, and do the data support the conclusions?

Reviewer #1: Yes

Reviewer #2: Yes

3. Has the statistical analysis been performed appropriately and rigorously? 

Reviewer #1: Yes

Reviewer #2: Yes

4. Have the authors made all data underlying the findings in their manuscript fully available?

Reviewer #1: Yes

Reviewer #2: Yes

5. Is the manuscript presented in an intelligible fashion and written in standard English?

Reviewer #1: Yes

Reviewer #2: Yes

6. Review Comments to the Author

Reviewer #1: The authors have taken care in the revision and have addressed all my original concerns. My only remaining comment: in revising the abstract, the conclusions section contains quite a bit of causal language, which should be avoided given that it's a correlational study. Apart from this I have no further comments and I don't need to see a revision. I look forward to seeing the paper in print at some stage soon.

All the best, Matthew

Response: We have now revised the Conclusions section in the Abstract so that there is no causal language in the sentences that relate to our results. We decided to keep the causal language in the last sentence, because that sentence is based on speculation about the meaning of the results. The text now reads as follows:

”Taken together, higher trait reactance seems to be relevant for attitudes and behaviors that go against conventional medicine, because trait reactance is connected to a distrust in medical doctors. Our findings also suggest that high trait reactance and low trust in doctors function differently for different people: For some individuals they might be associated with anti-vaccination attitudes and behavior, while for others they might be related to CAM use. We speculate that this is because people differ in what is important to them, leading them to react against different aspects of conventional medicine.”

Reviewer #2: Thanks for addressing my comments seriously. As well as being more polished overall, the revised analyses and results provide a much clearer and less ambiguous interpretation, which well supports the conclusions. An interesting result, and good work.

7. PLOS authors have the option to publish the peer review history of their article (what does this mean?). If published, this will include your full peer review and any attached files.

Do you want your identity to be public for this peer review? For information about this choice, including consent withdrawal, please see our Privacy Policy.

Reviewer #1: Yes: Matthew Hornsey

Reviewer #2: No

---

## [Editor Report · Decision Letter 2]

9 Jul 2020

Trait reactance and trust in doctors as predictors of vaccination behavior, vaccine attitudes, and use of complementary and alternative medicine in parents of young children

PONE-D-20-10006R2

Dear Dr. Soveri,

We’re pleased to inform you that your manuscript has been judged scientifically suitable for publication and will be formally accepted for publication once it meets all outstanding technical requirements.

Kind regards,

Peter Karl Jonason

Academic Editor

PLOS ONE
---

## [Editor Report · Acceptance letter]

13 Jul 2020

PONE-D-20-10006R2 

Trait reactance and trust in doctors as predictors of vaccination behavior, vaccine attitudes, and use of complementary and alternative medicine in parents of young children 

Dear Dr. Soveri:

I'm pleased to inform you that your manuscript has been deemed suitable for publication in PLOS ONE. Congratulations! Your manuscript is now with our production department. 

Kind regards, 

on behalf of

Dr. Peter Karl Jonason 

Academic Editor

PLOS ONE